# ENHANCING ZERO-SHOT TEXT-TO-SPEECH SYNTHESIS WITH HUMAN FEEDBACK

## ABSTRACT

In recent years, text-to-speech (TTS) technology has witnessed impressive advancements, particularly with large-scale training datasets, showcasing human-level speech quality and impressive zero-shot capabilities on unseen speakers. However, despite human subjective evaluations, such as the mean opinion score (MOS), remaining the gold standard for assessing the quality of synthetic speech, even state-of-the-art TTS approaches have kept human feedback isolated from training that resulted in mismatched training objectives and evaluation metrics. In this work, we investigate a novel topic of integrating subjective human evaluation into the TTS training loop. Inspired by the recent success of reinforcement learning from human feedback, we propose a comprehensive *sampling-annotating-learning* framework tailored to TTS optimization, namely **un**certainty-aware **o**ptimization (UNO). Specifically, UNO eliminates the need for a reward model or pairwise preference data by directly maximizing the utility of speech generations, while considering the uncertainty that lies in the inherent variability in subjective human speech perception and evaluations. Experimental results of both subjective and objective evaluations demonstrate that UNO considerably improves the zero-shot performance of TTS models in terms of MOS, word error rate, and speaker similarity. Additionally, we present a remarkable ability of UNO that it can adapt to the desired speaking style in emotional TTS seamlessly and flexibly.

## 1    INTRODUCTION

Artificial intelligence-generated content (AIGC) has attracted a surge of interest in both academia and industry, revolutionizing the way we acquire and generate information (Cao et al., 2023). In this context, learning from human feedback plays a pivotal role in calibration—it aims to align the generative models with human preferences. For instance, the reinforcement learning from human feedback (RLHF) technique can effectively help large language models (LLMs) to avoid generating harmful and toxic content (Achiam et al., 2023; Bai et al., 2022), which is crucial to the success of helpful systems like ChatGPT. Similar methods have recently been employed in text-to-image generation (Lee et al., 2023; Xu et al., 2024).

As an important task of AIGC, text-to-speech (TTS) synthesis technology is undergoing rapid development driven by deep learning models (Betker, 2023; Ren et al., 2020). Recent TTS works (Łajszczak et al., 2024; Ju et al., 2024; Peng et al., 2024; Wang et al., 2023a) with extensive text-speech training pairs exhibit remarkable zero-shot capacity which generates high-quality speech for speakers unseen during training. However, unlike the widespread application of RLHF in LLMs' calibration, aligning synthetic speech generation with human preferences remains challenging and has not yet been adopted in practice (Zhang et al., 2024). This contradicts the use of human subjective evaluation, such as the mean opinion score (MOS), as the gold standard for assessing TTS performance, and results in a clear mismatch between TTS training and evaluation. Motivated by this, we raise our basic research question: Can we integrate human feedback into the TTS learning loop?

The technical barrier of this topic stems from both sides of "TTS" and "human". (1) First, existing TTS models typically learn a mapping function from input word/phoneme sequences to either mel-spectrograms or discrete audio tokens followed by high-resolution waveform generation based on supervised training (Wang et al., 2023a). However, this learned mapping hardly provides diverse yet plausible generations based on the same text and speech prompt, which hinders the formulation of

pairwise preference data required by widely used LLMs alignment methods such as direct preference optimization (DPO) (Rafailov et al., 2024) (more discussion in Appendix B). (2) Second, human evaluations of speech quality are subjective and inherently personalised. Each individual's acoustic perception is unique and influenced by their physical state and cognitive biases, thereby resulting in *uncertainty* in the human feedback of each sample.

To address the above challenges, we propose a pioneering method named uncertainty-aware optimization (UNO), which aims to enhance zero-shot TTS performance with (simulated) human feedback. Inspired by the recent success of RLHF, UNO encompasses a *sampling-annotating-learning* pipeline, but its original design tailored to TTS lies in these three sub-steps:

- Sampling with diversity. To obtain representative training examples, UNO performs zero-shot TTS sampling with different speech prompts. It significantly contributes to the diversity in self-generated speech samples, thereby reducing the potential bias that arises from unrepresentative or skewed data collection.

- Annotating with uncertainty. UNO eliminates the dependency on preference data based on the same input. Instead, it allows a more flexible and tolerant data annotation: only a binary signal is required for whether the generated speech is desirable or not. Furthermore, since speech evaluation is subjective and personalised, this step encompasses the uncertainty caused by the individual differences among human annotators.

- UNO treats human feedback as a form of supervision with inconsistent labels to mitigate the mismatch between the TTS training objectives and MOS-like subjective evaluation metrics. This learning approach directly maximizes the utility of generations from TTS sampling, instead of relying on a reward model or maximizing the log-likelihood of preferences.

Experimental results show that UNO comprehensively enhances the performance of zero-shot TTS models, including speaker similarity (SIM), word error rate (WER), and pseudo-MOS estimated by three pre-trained models. For validation, we conduct subjective human listening tests in the form of naturalness MOS scoring and side-by-side A/B testing. The results of human evaluation confirm that the TTS model optimized through our method significantly outperforms the baseline ($3.38 \rightarrow 4.06$) and sounds equally good compared to the ground truth speech. Through both token-level and utterance-level visualization, it is observed that UNO provides effective supervised signals, resulting in the distribution of generated content being closer to the ground truth distribution. Furthermore, UNO exhibits flexible scalability through adjusting optimization objectives. By changing the selection criteria in sampling selection, the method can be seamlessly extended to emotional TTS.

Our contributions are summarised as follows: (1) We present a comprehensive framework tailored to zero-shot TTS optimization, where human feedback is taken into account in the training objectives to alleviate the mismatch between training and evaluation. (2) By delving into the characteristics of speech synthesis, UNO eliminates the dependence on preference data and accommodates the uncertainty in human subjective evaluations. This provides a new perspective for high-dimensional modality generation and alignment. (3) Intensive experiments demonstrate that UNO brings remarkable performance gain to zero-shot TTS models, especially in avoiding most failed cases. Additionally, UNO can be seamlessly extended to emotion TTS, demonstrating its scalability and practical value.

## 2 RELATED WORK

**Text-to-speech as language modelling.** Inspired by the success of LLMs (Brown et al., 2020), formatting TTS task as next token prediction has gained remarkable popularity in recent years (Borsos et al., 2023; Rubenstein et al., 2023; Zhang et al., 2023a). Under this setup, the prior step is to convert speech waveform into a sequence of learnable and discrete units based on vector quantization (Défossez et al., 2022; Zeghidour et al., 2021). SpeechTokenizer (Zhang et al., 2023b) and RepCodec (Huang et al., 2023) enhance the semantic tokenization by adding self-supervised embedding prediction-based losses (Mohamed et al., 2022). With discrete acoustic tokens, TortoiseTTS (Betker, 2023) pioneers to combine a speech-code language model with a diffusion decoder to achieve few-shot TTS. VALL-E (Wang et al., 2023a) and Spear-TTS (Kharitonov et al., 2023) scales to use 60k training data using a pre-trained neural codec model (Défossez et al., 2022), which exhibits remarkable zero-shot capacity to synthesize speech for unseen speakers with speech prompt. VALL-E X (Zhang et al., 2023c) and VioLa (Wang et al., 2023b) extent this framework to cross-lingual TTS, and later works (Lyth & King,

2024; Ji et al., 2024; Liu et al., 2023; Yang et al., 2023a) control the style of speech synthesis based on the neural codec. More recent work (Peng et al., 2024) extends the TTS framework to address speech editing tasks, BaseTTS (Łajszczak et al., 2024) built the first a billion-parameter TTS model based on a decoder-only structure. RALL-E (Xin et al., 2024) presents a robust language modeling approach for zero-shot TTS.

**Learning from Human Feedback.** Human feedback has been widely used in language models for NLP tasks, such as text translation (Kreutzer et al., 2018), instruction-following (Ouyang et al., 2022), and summarization (Stiennon et al., 2020). The advancements in the RLHF framework have contributed to training helpful and harmless AI agents aligning with human preference in the past years (Christiano et al., 2017; Bai et al., 2022; Achiam et al., 2023; Dai et al., 2023). Moreover, recent works like DPO (Rafailov et al., 2024) shift in favour of closed-form losses that directly operate on preference data. Different from typical preference-based RL (Jain et al., 2013; Busa-Fekete et al., 2014), DPO-style works remove the explicit reward model learning and provide the same alignment effect (Zhou et al., 2023; Amini et al., 2024; Zeng et al., 2024; Liu et al., 2024) with typical RLHF. Moreover, (Chen et al., 2024b; Yuan et al., 2024) propose to calibrate LLMs in a "self-rewarding" manner, and KTO (Ethayarajh et al., 2024) relieves the dependence on preference data and optimises LLMs using prospect theory (Kahneman & Tversky, 2013).

**Summary.** To align TTS systems with human preference, we propose to consider the *uncertainty* of human subjective evaluations (Maniati et al., 2022)—assessing speech has more perspectives and subjectivity than text (van Heuven & van Bezooijen, 1995; Wiebe et al., 2004). Moreover, UNO eliminates the dependence on pairwise preference data and does not require any corresponding ground truth speech (different from SpeechAlign (Zhang et al., 2024)). Combining these factors, we regard UNO as a step towards incorporating human evaluation signals in TTS training and contributing to developing more powerful and versatile speech synthesis.

## 3 BACKGROUND

**Neural Codec Language Modeling for TTS.** Speech synthesis aims to convert a sequence of transcript $t$ into a corresponding speech waveform $s$. This mapping can be formally expressed using a function $\pi$ parameterized by $\theta$ as $s = \pi_\theta(t)$. In this work, we regard the TTS problem as a conditional speech-codec-based language modelling task as proposed in (Wang et al., 2023a). $s \in \mathbb{R}^{L \times n}$ is tokenized into sequences of discrete acoustic units with a neural codec encoder, where $L$ is the downsampled utterance length and $n$ is the number of residual vector quantization (RVQ) codebooks. After quantization, the neural codec decoder is able to reconstruct the waveform.

**Zero-shot TTS** extends conventional TTS by enabling speech synthesis using voices unseen during model training. Given the target transcript $t$ and a short speech prompt $p$ as reference, zero-shot TTS is framed as a transcript-conditioned speech continuation task that uses well-trained model $\theta = \theta_1 \cup \theta_2$ to predict the first layer of codebook $s_L^{(1)}$ with corresponding content and speaker's voice using parameter $\theta_1$ in an *autoregressive* manner, and then predict other codebooks $s_L^{(2:n)}$ using parameter $\theta_2$ in a *non-autoregressive* manner. This hierarchical structure is denoted as:

$$s_l^{(1)} = \pi_{\theta_1}(p^{(1)}, s_{<l}^{(1)}, t),\ l \in \{1, 2, \ldots, L\} \tag{1}$$

$$s_L^{(n)} = \pi_{\theta_2}(p^{(1:n)}, s_L^{(1:n-1)}, t) \tag{2}$$

where $p$ can be viewed as a prefix sequence during decoding, and $s_{<t}$ is the history predicted by the model. Typically, the $s_L^{(1)}$ represents the acoustic properties like speech content, while $s_L^{(2:n)}$ recovers fine acoustic details.

**RLHF with Preference Data.** Given a dataset $\mathcal{D}$ with preference data point $(x, y_w, y_l)$, where $y_w$ and $y_l$ are the win-loss generations based on the same input $x$, it is assumed that the probability of $y_w$ is preferred to $y_l$ can be captured by a "true" reward function $R^*$. Since obtaining $R^*$ from humans would be intractably expensive, prior RLHF work employs a reward model $R_\phi$ as a proxy trained by minimizing the negative log-likelihood of the human-annotated data:

$$\mathcal{L}_{R_\phi} = \mathbb{E}_{x,y_w,y_l \sim \mathcal{D}}\left[-\log \sigma(R_\phi(x, y_w) - R_\phi(x, y_l))\right], \tag{3}$$

where $\sigma$ is the logistic function. Furthermore, a reference model $\pi_{\text{ref}}$ with KL divergence penalty is introduced to prevent the model $\pi_\theta$ from making radical update, the maximizing objective is:

$$\mathbb{E}_{x \in \mathcal{D}, y \in \pi_\theta}[R_\phi(x, y)] - \beta D_{\text{KL}}(\pi_\theta(y|x) \| \pi_{\text{ref}}(y|x)), \tag{4}$$

where $\beta$ is a balancing weight. Since the first item is non-differentiable for backpropagation, an RL algorithm like PPO is required to pursue maximum rewards and optimize the policy network $\pi_\theta$.

RLHF typically requires high computational costs for sampling generations and, additionally, exhibits instability in practice. Recent advances like DPO (Rafailov et al., 2024) focus on closed-form losses that present an implicit reward function under the RLHF objective in Eqn. equation 4, where the optimal reward for an input-output pair is denoted as:

$$R^*(x, y) = \beta \log \frac{\pi_\theta(y|x)}{\pi_{\text{ref}}(y|x)} + \beta \log Z(x), \tag{5}$$

where $Z(x)$ is a partition function. Then Eqn. equation 5 is utilized to maximize the margin between preferred and dispreferred samples, which has been demonstrated mathematical equivalence with typical KL-constrained RLHF in Eqn equation 4.

**Variability in Human Evaluations.** Variability is a unique aspect of real-world human evaluation. Individual variations in physical states, cognitive biases, and personal experiences can lead to subjectivity in perceptual quality assessment (*e.g.* TTS quality assessment). Instead of solely relying on mean opinions, we propose incorporating the variability present in human evaluation, which helps mitigate potential biases and promotes fairness and inclusivity. Prior approaches for modelling variability in human annotations can be broadly grouped into two types. The first approach explicitly models the behaviours of different annotators using different individual models (Fayek et al., 2016; Chou & Lee, 2019; Davani et al., 2022), which is not scalable when the number of annotators increases. The second approach approximates subjective probability distributions using Markov chain Monte Carlo with people (Sanborn & Griffiths, 2007; Harrison et al., 2020), which requires human annotators to be dynamically involved in the process. In this work, a meta-learning framework is adopted for zero-shot human annotation distribution estimation. Given a synthesized utterance $s_i$ and a set of $M_i$ human annotations $\mathcal{D}_i = \{y_i^{(m)}\}_{m=1}^{M_i}$ associated with $s_i$. The simulator aims to model the conditional annotation distribution $\mathrm{p}(y_i|s_i)$. For an unseen test utterance $s_*$, the simulator can then predict $\mathrm{p}(y_*|s_*)$ to simulate human-like annotations $\mathcal{D}_* = \{y_*^{(m)}\}_{m=1}^{M_*}$ in a way that reflects how it would be labeled by human annotators. The framework involves meta-learning a deep neural network model to estimate $\mathrm{p}(y_i|s_i)$ across all training data $\mathcal{D} = \{(s_i, \mathcal{D}_i)\}_{i=1}^N$ where $N$ is the number of training samples. The deep neural network model then serves as a distribution estimator to allow efficient generation of human-like annotations.

## 4 METHODOLOGY

### 4.1 DATA SAMPLING AND ANNOTATING

In order to acquire representative data, we introduce a simple sampling strategy that can encourage more diversified zero-shot TTS generation. Specifically, for each target transcript $x$, we sample a batch of speech prompts $\{p_1, p_2, \ldots, p_b\}$ with the size of $b$ from an unseen speaker pool. Then $x$ is alternately combined with $k$-th reference $p_k$ from the batch to form a different input to the TTS model by $s_k = \pi_\theta(t, p_k)$, $k \in \{1, 2, \ldots, b\}$. Note that $p_k$ acts as a prefix sequence in autoregressive decoding and significantly contributes to the diversity of target speech $s_k$.

After completing $b$ inferences for a batch, the desirable and undesirable samples are distinguished by human and respectively stored in $\mathcal{P}_{\text{pos}}$ and $\mathcal{P}_{\text{neg}}$ pools. Moreover, since human evaluations of generated speech are less intuitive than text, we record the *uncertainty* $u$ associated with each data point. When multiple annotators provide distinct decisions, $u$ can be the variance of evaluations for this assessment. We finally obtain two pools with $I$ desirable and $J$ undesirable samples after sampling $K$ times, respectively:

$$\mathcal{P}_{\text{pos}} = \{(t_i, p_i, s_i; u_i) \mid s_i \sim \pi_{\text{ref}}(t_i, p_i), u_i \in [0, 1), i = 1, 2, \ldots, I\} \tag{6}$$

$$\mathcal{P}_{\text{neg}} = \{(t_j, p_j, s_j; u_i) \mid s_i \sim \pi_{\text{ref}}(t_j, p_j), u_i \in [0, 1), j = 1, 2, \ldots, J\} \tag{7}$$

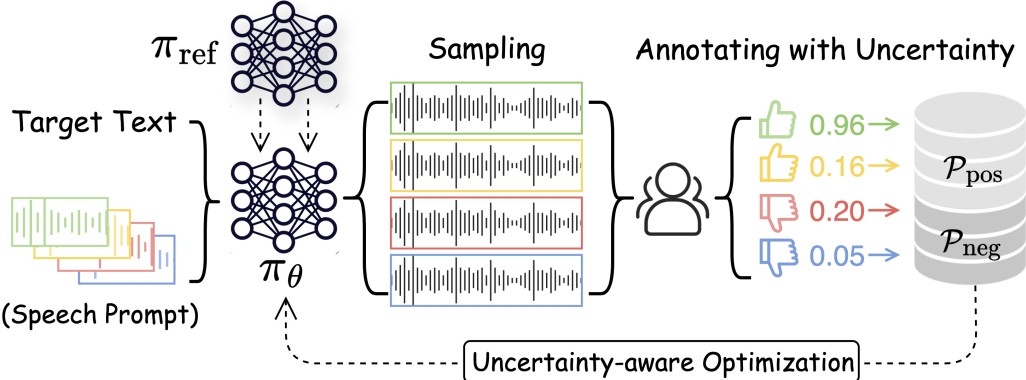

Figure 1: This sampling-annotating-learning framework of UNO. In annotating, the "like" and "dislike" symbols denote the binary signal for whether this synthetic speech is desirable or not, and the digits represents the uncertainty caused by the variability of annotators.

where $I$ and $J$ may not be equal. This indicates that the samples in $\mathcal{P}_{pos}$ and $\mathcal{P}_{neg}$ are *not* pairwise preference data since they are based on diversified speech prompts. This significantly helps to increase the diversity of generated speech (see more analysis in Appendix C), thus potentially reducing the bias of data collection for the subsequent optimization process.

Considering the substantial human resource consumption required in annotations, this paper utilizes anthropomorphic annotation simulators trained with real human-labeled SOMOS dataset (Maniati et al., 2022) for efficient generation of evaluation labels while simulating variability in human opinions. Both a discriminative simulator, EDL (Wu et al., 2023b), and a generative simulator, I-CNF (Wu et al., 2023a), are used to simulate the human decision with uncertainty. EDL makes a Gaussian assumption on the conditional annotation distribution $p(y|s)$ and places a normal inverse-gamma (NIG) over the Gaussian likelihood to learn a higher-order prior distribution, also called the evidential distribution (Amini et al., 2020):

$$\{y^{(m)}\}_{m=1}^M \sim \mathcal{N}(\mu, \sigma^2), \quad \mu \sim \mathcal{N}(\gamma, \sigma^2 v^{-1}), \quad \sigma^2 \sim \Gamma^{-1}(\alpha, \beta). \tag{8}$$

A deep neural network model is trained to predict the hyperparameters $\Omega = \{\gamma, v, \alpha, \beta\}$ of the NIG prior by maximizing the marginal likelihood of sampling from all possible Gaussians:

$$p(y|\Omega) = \int p(y|\Psi)p(\Psi|\Omega)d\Psi = \text{St}_{2\alpha}\left(y|\gamma, \frac{\beta(1+v)}{v\,\alpha}\right), \tag{9}$$

where $\Psi = \{\mu, \sigma\}$ is the hyperparameters of the Gaussian likelihood and $\text{St}_\nu(t|r, s)$ is the Student's t-distribution evaluated at $t$ with location parameter $r$, scale parameter $s$, and $\nu$ degrees of freedom. The predicted mean and variance can be computed analytically as $\mathbb{E}[y] = \gamma$ and $\text{Var}[y] = \beta(1+v)/v(\alpha-1)$. The predicted variance is then used as uncertainty. I-CNF removes the Gaussian assumption of EDL by meta-learning a conditional normalizing flow $p_\phi(y|s) = \int p_\phi(y|z)p_\Lambda(z|s)dz$ where $z$ is a latent variable sampled from a Gaussian distribution conditioned on input $s$. The mean and variance of the conditional Gaussian prior are parameterized by a neural network model with parameters $\Lambda$ as $p_\Lambda(z|s) = \mathcal{N}(z|\mu_\Lambda(s), \text{diag}(\sigma_\Lambda^2(s)))$. The simulated evaluation $y$ is obtained by a deterministic invertible transformation $p_\phi(y|z) = \delta(y - f_\phi(z))$, where $f_\phi(z)$ is parameterized by an invertible neural network model $\phi$, and $\delta(\cdot)$ is the multivariate Dirac delta function. That is,

$$p(y|s) = \int \delta(y - f_\phi(z))p_\Lambda(z|s)dz = p_\Lambda\left(f_\phi^{-1}(y)\Big|s\right)\left|\det\left(\frac{\partial f_\phi^{-1}(y)}{\partial y}\right)\right|, \tag{10}$$

where $\det(\cdot)$ denotes the determinant operator, $\partial f_\phi^{-1}(y)/\partial y$ denotes the Jacobian matrix of $f_\phi^{-1}(y)$. This modelling choice has the advantage of having tractable marginal likelihood as in Eqn. equation 10 while not restricting the intermediate variable $y$ to a specific type of distribution. At test time, the I-CNF can simulate human-like annotations for an unseen, unlabeled descriptor $s_*$ by first drawing $\{z_*^{(m)}\}_{m=1}^{M_*} \sim p_\Lambda(z|s_*)$ from the conditional prior, then applying transformation $y_*^{(m)} = f_\phi(z_*^{(m)})$. The uncertainty can be computed as the variance of $\{y_*^{(m)}\}_{m=1}^{M_*}$. Since the sampling process can be batch processed, I-CNF thus allows efficient simulation of human evaluations.

### 4.2 UNCERTAINTY-AWARE LEARNING FOR TTS

**Why is DPO Eliminated?** To optimize the KL-constrained RLHF objective given in Eqn. equation 4, DPO (Rafailov et al., 2024) presents an approach with mathematical equivalence by maximizing the margin between the preferred and unpreferred generations based on the same input. Especially, the training criterion can be written as follows under the TTS formulation:

$$\mathcal{L}_{\text{DPO-TTS}}(\pi_\theta, \pi_{\text{ref}}) = \mathbb{E}\left[ -\log \sigma \left( \beta \log \frac{\pi_\theta(s_w|t, v)}{\pi_{\text{ref}}(s_w|t, v)} - \beta \log \frac{\pi_\theta(s_l|t, v)}{\pi_{\text{ref}}(s_l|t, v)} \right) \right], \quad (11)$$

where $s_w$ and $s_l$ are supposed to be preferred and unpreferred speech obtained using the same transcript and speech prompt $(t, v)$. The sampling strategy introduced in Sec. 4.1 fails to provide pairwise $s_w$ and $s_l$ required by Eqn. equation 11, as the data point in both $\mathcal{P}_{\text{pos}}$ and $\mathcal{P}_{\text{neg}}$ are generated using different speech prompts. In fact, due to the lack of diversity, existing TTS models struggle to generate paired $s_w$ and $s_l$ using fixed target transcripts and speech prompts. A solution recently proposed in (Zhang et al., 2024) recalls ground truth speech as $s_w$ while treating the generated speech by the model as $s_l$. A difficulty of this approach lies in the fact that the TTS model's outputs are not necessarily unpreferred, and the ground truth may not always be accessible in practice. Another solution is adjusting the sampling hyper-parameters or using beam search strategy. However, it results in low data efficiency, and is exposed to less unseen speakers compared with same sampling times.

**Uncertainty-aware Optimization.** To remove the dependence on preference data, a promising solution is to anchor a "reference point" that is added or subtracted to get the relative gain or loss respectively. To this end, we utilize the KL term $Z_{\text{ref}}$ introduced in KTO (Ethayarajh et al., 2024) that is defined as:

$$Z_{\text{ref}} = \mathbb{E}_{(t', v', s') \sim \mathcal{P}_{\text{pos}} \cup \mathcal{P}_{\text{neg}}}[\text{KL}(\pi_\theta(s'|t', p') \| \pi_{\text{ref}}(s'|t', p'))] \quad (12)$$

where $(t', v', s')$ samples from each batch during training. $Z_{\text{ref}}$ is not involved in the backpropagation process, while it makes the training more stable like the role of the *baseline* in REINFORCE (Sutton & Barto, 2018). With the existence of $Z_{\text{ref}}$, we can directly maximizes the utility of generations from $\mathcal{P}_{\text{pos}}$ and $\mathcal{P}_{\text{neg}}$ as follows using a value function $V_{\text{TTS}}$ with logistic function $\sigma$:

$$V_{\text{TTS}}(t, p, s; u) = \begin{cases} \sigma(u^{-1} \cdot R(t, p, s) - Z_{\text{ref}}), & \text{if } (t, p, s; u) \sim \mathcal{P}_{\text{pos}} \\ \sigma(Z_{\text{ref}} - u^{-1} \cdot R(t, p, s)), & \text{if } (t, p, s; u) \sim \mathcal{P}_{\text{neg}} \end{cases} \quad (13)$$

$$R(t, p, s) = \log \frac{\pi_\theta(s|t, p)}{\pi_{\text{ref}}(s|t, p)} \quad (14)$$

where $R(t, p, s)$ is the implicit reward modeling under RLHF objective in Eqn. equation 4 and normalized inverse uncertainty $u^{-1}$ controls the magnitude of model $\pi_\theta$ updates from $\pi_{\text{ref}}$, replacing the original hyper-parameter $\beta$ in DPO. The motivation behind this design is that reward allocation should consider the uncertainty in human feedback associated with the sample. Intuitively, the model is allowed to update more aggressively given a desirable generation with low uncertainty, and conversely, the updates are more conservative when there is high uncertainty. Based on this, the optimization loss is written as:

$$\mathcal{L}_{\text{TTS}}(\pi_\theta, \pi_{\text{ref}}) = \mathbb{E}_{t, p, s \sim \mathcal{P}_{\text{pos}} \cup \mathcal{P}_{\text{neg}}}(1 - V_{\text{TTS}}(t, p, s; u)). \quad (15)$$

If $s$ is a desirable data point sampled from $\mathcal{P}_{\text{pos}}$, then the probability of $\pi_\theta$ is boosted to minimize the loss, but the $Z_{\text{ref}}$ also increases. This forces the model to learn exactly what makes an output desirable without dispensable update based on $\pi_{\text{ref}}$.

## 5 EXPERIMENTS SETUP

**TTS Data.** The data used in our experiments includes three parts: supervised pre-training for the TTS model, optimization with UNO, and evaluation. There are no overlapping speakers between them. (1) The GigaSpeech dataset (Chen et al., 2021) is used as training data to train the supervised TTS model from scratch, which contains 9k hours of audiobooks, podcasts, and YouTube videos at a 16kHz audio sampling rate. (2) The LibriTTS (Zen et al., 2019) dataset which has no overlapping with Gigaspeech is used for UNO. More specifically, we sample a pool of speech prompts consisting of audio files around 3 seconds (commonly used in zero-shot TTS studies), and then perform zero-shot

TTS generation based on other target transcripts of more than 6 words. Notably, this process does not require the ground-truth speech of the target transcript. (3) For evaluation, we use a subset from LibriSpeech test-clean (Panayotov et al., 2015) with the audio lengths between 4 and 10 seconds (keeping consistency with (Chen et al., 2021)), and select the 3-second audio files as speech prompt according to their speaker identities.

**Models.** We employ VoiceCraft (Peng et al., 2024) as the baseline model due to its demonstrated superior zero-shot TTS capability, where both base (330M) and large (830M) pre-trained models are considered as the starting points in subsequent experiments. Speech Tokenizer is the pre-trained Encodec with 4 RVQ codebooks and a vocabulary of size 2048. More details are introduced in the Appendix D.

**Objective Evaluation.** Following prior studies, the metrics of WER and SIM are used in this work, which are calculated using pre-trained Whisper-medium.en and WavLM-TDCNN speech and speaker recognition models respectively. Furthermore, we use the *MOSNet* to estimate an objective MOS for reference, which is reported to have good generalization capability to out-of-domain data.

**Human Evaluation.** We randomly sample 40 listening examples from 4-6 seconds, 6-8 seconds, and 8-10 seconds, respectively, in order to cover different length of generations. Then, these 120 synthetic speech samples are assessed by ten listeners for naturalness MOS evaluation. Listeners were tasked with rating the naturalness of each audio sample on a 5-point Likert scale, ranging from 1 (very unnatural) to 5 (completely natural). Furthermore, these 120 speech samples are randomly assigned to ten listeners for side-by-side A/B testing (12 samples per person). After listening to two samples with the same speech content, listeners were asked to decide which one sounded more natural, or if they were too close to call, indicating a tie.

**Baselines.** In addition to the well-trained *VoiceCraft* model by typical supervised learning, we reproduce the following optimization approaches based on *VoiceCraft* system for comparison:

- *SpeechAlign-DPO*: Proposed by (Rafailov et al., 2024), it adapts the DPO algorithm to the TTS task and achieves better performance than other alignment methods.

- *SpeechAlign-ODPO*: (Amini et al., 2024) presents a enhanced version of DPO with considering offset. We use the difference between the estimated MOS of ground truth and the MOS of generated speech as "offset" to achieve ODPO optimization.

- *PPO-SDP*: We apply PPO optimization by directly employing MOSNet as the reward model and the mean of MOS as the reward signal. Furthermore, as standard deviation is available, we implement the Standard Deviation-Based Penalty method proposed in (Yang et al., 2024).

- *GroundTruth*: Since ground truth waveforms of the evaluation set are available, we calculate their corresponding metrics for TTS reference.

Notably, *SpeechAlign-DPO* and *SpeechAlign-ODPO* require ground truth to serve as positive samples ($y_w$), thus resulting in an unfair comparison with our approach.

## 6 RESULT AND ANALYSIS

### 6.1 OBJECTIVE RESULTS.

We report the objective results in Table 1. *I-CNF* and *EDL* models are recalled for MOS estimation as a reference, and *MOSNet* is a detached evaluator since it is not involved in optimization. From Table 1, we observe that (1) Both *UNO-ICNF* and *UNO-EDL* significantly enhance the TTS performance of *VoiceCraft* in terms of WER, SIM, and all estimated MOS, even approaching the corresponding results of GroundTruth. *I-CNF* and *EDL* tend to predict lower scores than *MOSNet* due to the data imbalance in their training SOMOS dataset. (2) SpeechAlign with preference data also enhances the baseline, and it avoids the human annotation process while relying on the ground truth speech during optimization. Compared with UNO, it views all synthetic speech as negative samples, possibly suppressing those outstanding generations. (3) The standard penalty benefits the stability of PPO. Using *I-CNF* and *EDL* as reward models, it surpasses the baseline without ground truth speech. The

Table 1: Objective results on WER (%), SIM, and MOS. "*Label*" denotes whether the approach requires labeled text-speech pairs ("✗" stands for label-free). For MOS evaluation, the "*ICNF*" and "*EDL*" are the models to estimate uncertainty during annotating, while "*MOSNet*" provide detached MOS estimation as it is not involved in the optimization process. The best results are in bold.

| Model | *Label* | WER↓ (%) | SIM↑ (0,1) | MOS ↑ *by* I-CNF | EDL | MOSNet |
|---|---|---|---|---|---|---|
| *VoiceCraft (baseline)* | - | 11.4 | 0.84 | 3.51 | 3.55 | 3.65 |
| *SpeechAlign-DPO* | ✓ | 7.2 | 0.91 | $3.70_{+0.19}$ | $3.72_{+0.17}$ | $3.86_{+0.21}$ |
| *SpeechAlign-ODPO* | ✓ | 6.9 | 0.90 | $3.73_{+0.21}$ | $3.76_{+0.24}$ | $3.90_{+0.25}$ |
| *PPO-SDP* | ✗ | 7.7 | 0.88 | $3.65_{+0.14}$ | $3.69_{+0.14}$ | $3.85_{+0.20}$ |
| *UNO-ICNF* | ✗ | 2.6 | 0.91 | $\mathbf{3.93}_{+0.42}$ | $3.90_{+0.35}$ | $\mathbf{4.31}_{+0.66}$ |
| *UNO-EDL* | ✗ | **2.4** | **0.92** | $3.88_{+0.37}$ | $\mathbf{3.91}_{+0.36}$ | $4.28_{+0.63}$ |
| *GroundTruth (upper-bound)* | - | 2.0 | - | 4.15 | 4.19 | 4.52 |

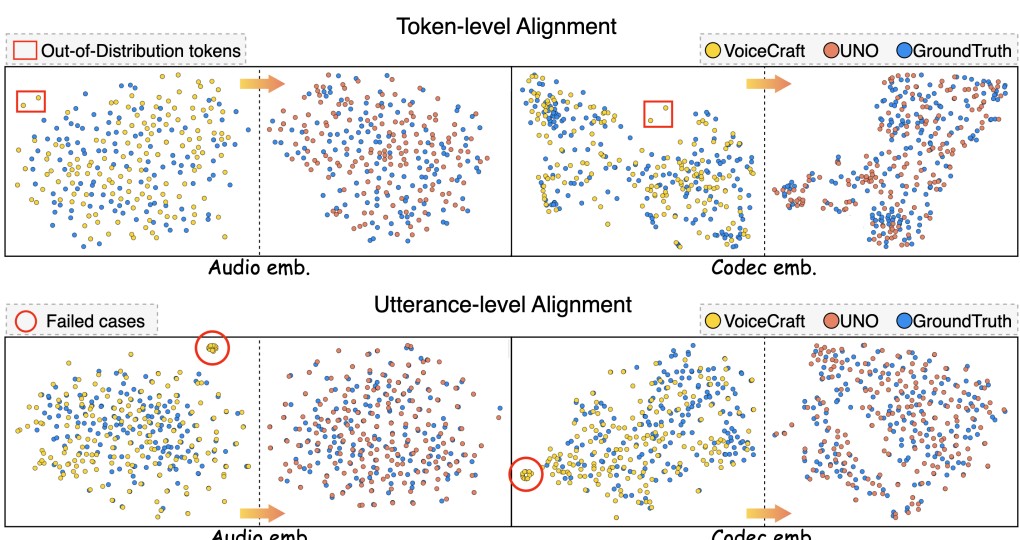

Figure 2: Visualization of UNO. The yellow-to-red arrow indicates the change before and after UNO. The token-level visualization (upper part) is projected by the generated tokens, while in utterance-level visualization (lower part), each point is projected by the embedding of an utterance. A cluster of data points shown in red circles are failed zero-shot TTS cases.

results on more zero-shot TTS system is Table 5. Additionally, we attach some listening examples on our demonstration page [1].

**Visualization of Optimization.** We show the effect of UNO through both token-level and utterance-level visualization in Figure 2. Specifically, both "Audio embedding" and "Codec embedding" of generated speech are projected into the same space by t-SNE. The former is extracted by the embedding layer of the TTS model that focuses on semantic information, and the latter merges all RVQ embeddings from the codec encoder that contains both speech content and acoustic details. Figure 2 shows that the red data points fit closer to blue data points than yellows, which indicates the UNO aligns the distribution of generative speech to ground truth speech. Furthermore, a cluster of yellow data points appears in utterance-level *VoiceCraft* (shown in red circles), representing *failed cases* of zero-shot TTS. To quantize, there are 18.5% generated speech that illustrates a WER higher than 15%, which we define as bad case. However, UNO removes this cluster of failure with only 4.9% bad cases remaining, which explains the source of improvements brought by our proposed UNO and indirectly reflects UNO's ability to improve the *robustness* of zero-shot TTS systems.

---

[1]https://uno-tts.github.io/listening-examples/

Table 2: Results on human evaluation.

| Model | MOS *by* | |
| --- | --- | --- |
| | Human | MOSNet |
| *VoiceCraft* | 3.38 | 3.57 |
| *UNO-ICNF* | 4.06 | 4.20 |
| *UNO-Human* | 3.98 | 4.13 |
| *GroundTruth* | 4.55 | 4.46 |

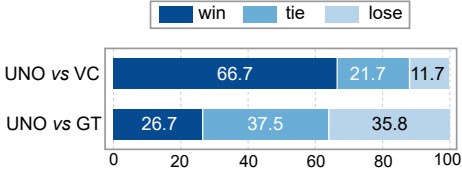

Figure 3: Result of A/B test. "VC" and "GT" denote the "VoiceCraft" and "GroundTruth".

Table 3: Result on emotional TTS in terms of Valence and Arousal attributes. "$\bar{v}$", "$\bar{a}$", and "$\bar{m}$" stand for the mean values of valence, arousal and MOS in each pool, respectively.

| EmotionTTS-Valence | | | | | | EmotionTTS-Arousal | | | | | |
| --- | --- | --- | --- | --- | --- | --- | --- | --- | --- | --- | --- |
| $\mathcal{P}_{pos}$ | | $\mathcal{P}_{neg}$ | | *Valence* ↑ | | $\mathcal{P}_{pos}$ | | $\mathcal{P}_{neg}$ | | *Arousal* ↑ | |
| $\bar{v}$ | $\bar{m}$ | $\bar{v}$ | $\bar{m}$ | *before* | *after* | $\bar{a}$ | $\bar{m}$ | $\bar{a}$ | $\bar{m}$ | *before* | *after* |
| 0.65 | 4.08 | 0.36 | 4.04 | 0.55 | 0.67 | 0.69 | 4.05 | 0.48 | 4.20 | 0.62 | 0.71 |

## 6.2 HUMAN EVALUATION.

We conduct both naturalness MOS scoring and A/B testing by the human listener to verify the performance improvements and report the MOS results in Table 2. Human evaluations overall are close to those of *MOSNet*, showing the efficacy of UNO. More importantly, we observe that UNO can considerably improve TTS robustness by avoiding most failed cases. Furthermore, we added a control group "*UNO-Human*" where humans perform both annotation (originally by *ICNF* and *EDL*) and evaluation, more details are in Appendix E. The MOS results show that UNO effectively aligns TTS with these annotator's preferences.

## 6.3 ANALYSIS ON UNCERTAINTY.

To examine the efficacy of uncertainty, we conduct an ablation study by establishing a baseline without uncertainty estimation, namely *UNO-null*. The variable $1/u$ in Eqn. equation 13 is replaced with a constant value of average uncertainty. The uncertainty and MOS results are reported in Table 4. It is observed that UNO achieves comparable MOS with *UNO-null*, but significantly reduces the variance on evaluation set compared with both *VoiceCraft* and *UNO-null*, even lower than GroundTruth. This indicates that UNO optimizes the generative speech towards consistency by different annotators.

We further test UNO on the 830M version of *VoiceCraft*, which is the largest open-source zero-shot TTS model up to now. As shown in Figure 4, UNO can also enhance the performance in terms of WER (2.7 → 2.2) and MOS (4.30 → 4.41). We also calculate the bad case (i.e., WER > 15%) ratio. Though *VoiceCraft-830M* indicates promising performance with 2.7% WER, it still produces 4.6% bad cases. In comparison, our proposed UNO can reduce the ratio to 1.9% and produce better performance as shown in Table 5, which indicates better robustness.

Table 4: Comparison results of uncertainty and MOS. $u^2$ is estimated by *I-CNF* models.

| Model | Unc. ($u^2$) | MOS |
| --- | --- | --- |
| *VoiceCraft* | 1.85 | 3.65 |
| *UNO-null* | $1.59_{-0.26}$ | $4.24_{+0.59}$ |
| *UNO-ICNF* | $1.34_{-0.51}$ | $4.31_{+0.66}$ |
| GroundTruth | 1.56 | 4.52 |

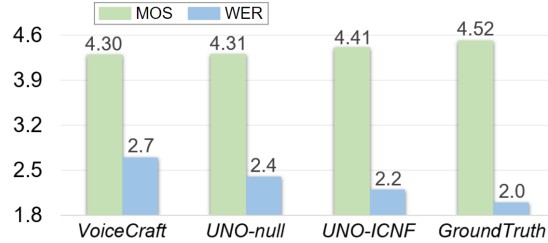

Figure 4: WER and MOS Results on 830M models.

### 6.4 EXTENSION ON REVERSE INFERENCE.

Due to the superior performance of the 830M model, we increase the sample size to obtain enough negative samples. At the same time, we face the issue of selecting positive samples, as the large number of positive examples lack distinctiveness. To address this, we propose a selection technique called reverse inference (UNO-RI) based on Bayes' theorem. We use the generated positive samples as prompts to synthesize the original prompt speech. This approach is used to test whether the synthesized speech can be input for a second inference, similar to the real speech. The results show that only 54% of the positive samples can produce synthesized speech with a MOS score above 3 when used as prompt speech. By adopting this positive sample selection strategy, we find that UNO-RI further improves the performance of the 830M model, achieving WER of 2.0 and MOS of 4.55, with failed cases reduced to less than 1%.

This results on UNO-RI provide two insights on zero-shot TTS. (i) While the synthesized speech may sound indistinguishable from real speech, when used as prompt speech input in a zero-shot TTS system, a noticeable gap emerges between the two. This suggests that their audio codec distributions still differ, and RLHF method can align the distribution of synthesized speech closed to real speech. (ii) The remarkable results of UNO-RI demonstrate that in preference optimization, the model is better suited than humans listener for selecting speech samples. Some highly fine-grained differences in speech are imperceptible to the auditory system of human. This stands in clear contrast to the approach of RLHF in the NLP domain.

### 6.5 EXTENSION ON EMOTIONAL TTS.

In addition to MOS, we extend UNO to align with other human preferences, allowing for the customization of TTS synthesis in different emotions. In practice, we establish the objective of optimization for emotional TTS by manipulating samples in the $\mathcal{P}_{pos}$ and $\mathcal{P}_{neg}$ based on emotional state model (Mehrabian, 1995).

**Valence.** We first utilize valence $v \in (0, 1)$ as a metric to prompt the TTS model to generate speech with *pleasant* emotion, which is equivalent to maximizing the valence in generations while keeping high MOS $m$ to ensure the quality. Specifically, our corresponding experimental adjustment consists of two parts. (1) We sample speech prompts from "happy (high $v$)" and "sad (low $v$)" categories from an emotional ESD dataset (Zhou et al., 2021) to encourage the diversity of $v$ in sampling generations. (2) We feed the samples with high valence and high MOS $(v+, m+)$ into $\mathcal{P}_{pos}$, and samples with low valence and high MOS $(v-, m+)$ into $\mathcal{P}_{neg}$. More details are illustrated in the Appendix F. The corresponding statistics for $v$ and $m$ are reported in the left part of Table 3, and the evaluation result based on "happy" prompts shows that UNO effectively achieves an absolute improvement of 0.12 $(0.55 \rightarrow 0.67)$. Surprisingly, 0.67 is even higher than the average $\bar{v}$ in $\mathcal{P}_{pos}$ (0.65), which shows UNO captures the desirable speech style to align with human preference.

**Arousal.** We also utilize arousal $\bar{a}$ to guide model generate speech with *surprise* emotion where speech prompts are samples from the "surprise" and "neural" categories of ESD dataset. Since they are not in opposite emotions, the average $\bar{a}$ in $\mathcal{P}_{neg}$ is only 0.48. However, UNO effectively improves the $a$ from 0.62 to 0.71 for evaluation, as shown in the right part of Table 3.

## 7 CONCLUSION

This paper presents a novel optimization method UNO tailored to zero-shot TTS models. UNO effectively integrates human feedback into the TTS learning objective using hundreds of self-generated samples, which are annotated by deep neural network models with desirable/undesirable pseudo labels and their corresponding label uncertainty. The subsequent optimization directly maximizes the utilization of these samples in an uncertainty-aware manner. Experimental results demonstrate the remarkable efficacy of UNO in terms of both objective metrics and subjective metrics scored by human evaluation. Meanwhile, we explore the extension of UNO on reverse inference and the emotional TTS tasks. We believe this work can provide unique insights and inspiration for leveraging human feedback to enhance the high-dimensional data generation performance of AIGC.

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

## A    FREQUENTLY ASKED QUESTIONS

***Question1****: What is the limitation of this work?*

Though UNO demonstrates promising efficacy of the neural codec TTS model, it is important to note that TTS is a highly prevalent task encompassing a wide range of different architectures. For instance, models with zero-shot TTS capability, such as those based on the probabilistic diffusion model, have also demonstrated remarkable performance of speech synthesis (Ju et al., 2024). Therefore, we may not have fully captured the potential versatility and broader applicability of human feedback in TTS optimization. However, in the realm of image generation, recent studies have shown that conditional diffusion models can also be aligned with human preferences using direct preference optimization (Wallace et al., 2023; Yang et al., 2023b), where the Eqn (11) can be applied in diffusion model training. Thus, it is theoretically feasible for UNO to optimize diffusion-based TTS models as well, and the first relevant work is proposed in (Chen et al., 2024a).

***Question2****: Why UNO can avoid the failed cases of original VoiceCraft?*

The failed cases in *VoiceCraft* mostly stem from the autoregressive generation property in TTS inference, where the TTS model does not learn a strict alignment between speech and text during its training. However, such failed cases are not stubborn and can be avoided by multiple sampling or CoT-like technique introduced in (Xin et al., 2024). In UNO, the $\mathcal{P}_{neg}$ covers different kinds of failed cases, and their corresponding implicit reward is suppressed during optimization. In human evaluation, listeners found that UNO can avoid all failed cases of repetition and interruption that happened in baseline, only remaining with some small errors such as mispronunciation or omission.

***Question3****: Can UNO be applied to other zero-shot TTS models?*

The results in Table 5 demonstrate the effectiveness of our proposed UNO on multiple popular zero-shot TTS models, including VALL-E[2] (Wang et al., 2023a), GPT-SoVITS[3], VoiceCraft-330M,

---

[2]`https://huggingface.co/amphion/valle_librilight_6k`
[3]`https://github.com/RVC-Boss/GPT-SoVITS`

Table 5: Performance of proposed UNO optimization method on more zero-shot TTS backbones.

| Model | WER↓ | SIM↑ | MOS ↑ |
|---|---|---|---|
| *VALL-E* | 28.4 | 0.68 | 3.03 |
| *VALL-E-UNO* | 15.8 | 0.80 | 3.45 |
| *GPT-SoVITS* | 12.6 | 0.78 | 3.34 |
| *GPT-SoVITS-UNO* | 5.0 | 0.88 | 4.06 |
| *VoiceCraft-330M* | 11.4 | 0.84 | 3.65 |
| *VoiceCraft-330M-UNO* | 2.6 | 0.91 | 4.31 |
| *VoiceCraft-830M* | 2.7 | 0.91 | 4.30 |
| *VoiceCraft-830M-UNO* | **2.2** | **0.93** | **4.41** |
| *Ground-Truth (upper bound)* | 2.0 | - | 4.52 |

and VoiceCraft-830M. Specifically, UNO consistently produces better performance over all metrics including WER, SIM and MOS, which indicates the generality of our proposed approach on different TTS backbones. In addition, the corresponding bad case ratios are also significantly improved as introduced in Section 6.3.

***Question4****: Can UNO be applied to other audio generation tasks like music generation?*

Yes, human preference is an important topic in music generation based on text description (Agostinelli et al., 2023) or text-to-audio generation (Liao et al., 2024). However, since it does not need strict token-level alignment like TTS, it is easier to collect pairwise data from humans, e.g., MusicRL (Cideron et al., 2024) propose a human-annotated dataset including 300,000 pairwise preferences. Instead of direct preference optimization, they use this dataset to train a reward model for music model optimization. Considering the consistent optimization objective, UNO can be used in music generation that directly maximizes the utility of music generation, more importantly, the subjective evaluation of generative music also exhibits variability caused by the listener's taste and perception. Therefore, the utilization of uncertainty in UNO potentially provides a solution to address evaluation variability in music generation.

***Question5****: How about other training strategies for UNO, such as tuning auto-regressive only?*

We conduct comparative experiments on different training approaches, including training autoregressive only, and non-autoregressive only, as well as LoRA tuning on 830M models. However, there is negligible impact on the final results. The underlying cause stems from the constraints of the reference model, which prevent over-fitting problems. Additionally, all training samples are generated by the model itself, which does not force the model to adapt to new data distributions.

***Question6****: Can UNO handle the data imbalance in $\mathcal{P}_{pos}$ and $\mathcal{P}_{neg}$?*

Yes, UNO can handle the case when $I$ ($\mathcal{P}_{pos}$) and $J$ ($\mathcal{P}_{neg}$) are imbalance. Specifically, when the ratio of positive to negative samples is set to 1:4 ($I = 50$, $J = 200$), the MOS performance only decreased by 0.11 ($4.31 \rightarrow 4.20$), while when the ratio is set to 4:1 ($I = 200$, $J = 50$), the MOS decreased by 0.30 ($4.31 \rightarrow 4.01$), but still significantly surpasses baseline (3.65). This is because, for a 330m model, a sufficient number of samples are required to cover failed cases. Furthermore, UNO does not rely on large amount of training data, we increase the $K$ to 10k ($I$ and $J$ are both 5k) but the performance gain is less than 0.1 ($4.31 \rightarrow 4.40$ by *MOSNet*) without hyper-parameter tuning. Since each sample should have been annotated by human, we set the $K$ to a few hundred to ensure that it is easy to implement in practice.

## B    MORE DISCUSSION ON LLM AND TTS CALIBRATION

LLMs have exhibited outperforming capacity in language generation. We first briefly introduce their calibration process as shown in the left part of Figure 5. A well-trained LLM is able to generate various responses based on the same prompt "*Mamba is*". These responses are diverse from different perspectives, but they are all reasonable and consistent with prompt input. In this case, human

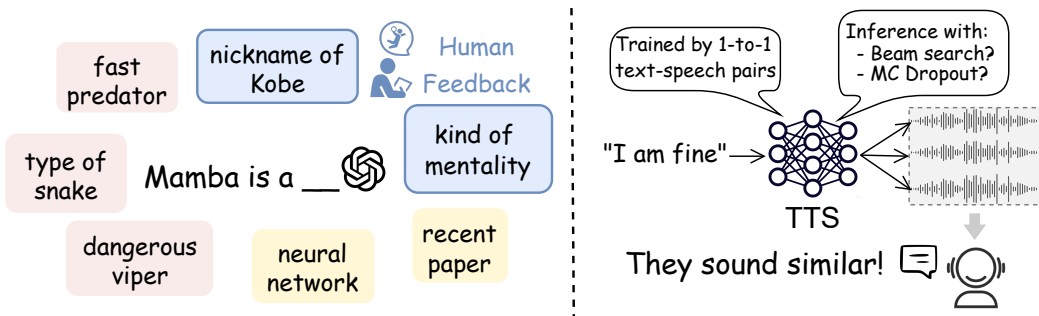

Figure 5: Comparison between LLM and TTS alignment.

annotators can label the responses of blue boxes as their preference if they plan to deploy this LLM into the basketball domain, and others (in yellow and pink boxes) are viewed as dispreferred samples to be filtered out. Thereafter, this preference data can be utilized to train a reward model for PPO, or directly optimize the LLM by DPO, thus aligning the generative content of LLMs with human preference by a "wide-to-narrow" calibration.

Recent TTS research mostly concentrates on enhancing the model's zero-shot capacity: given a short reference recording of the unseen voice (a.k.a, speech prompt), zero-shot TTS is framed as a transcript-conditioned speech continuation task, where the synthetic speech is expected to keep consistent with speech prompt. However, unlike humans who can speak a transcript with various styles, existing TTS models struggle to generate diverse speech (e.g., speed, prosody, emotion, etc) based on the same transcript and speech prompt. Beam search and Monte-Carlo dropout can introduce randomness into neural codec modelling TTS system, however, humans can not easily distinguish preference for speech generations like text, as shown in the right part of Figure 5. Therefore, considering the significant impact of the speech prompt on generated speech, this work directly varies the speech prompt during zero-shot TTS sampling. Though this approach hinders the formulation of pairwise preference data based on the same input, it highly encourages diversity in generated speech, thereby reducing the bias in data collection and benefiting the subsequent optimization process.

In general, we summarize the difference between designing a TTS optimization method and typical LLMs-based RLHF: (i) RLHF mostly serves as a role of calibrator in LLMs generation. However, TTS optimization requires learning from human evaluation to mitigate the absence of such supervised information during training. (ii) Compared with LLMs, TTS systems fail to produce diverse and representative samples based on the same input, thus TTS optimization requires eliminating the dependence of pairwise preference data. (iii) Subjective evaluation of synthetic speech is not as straightforward as text. For instance, it is hard to judge if one synthetic speech misses some words but another is at an unnatural pace, but it frequently happens in zero-shot TTS.

## C  DISCUSSION ON DATA SAMPLING

We first visualize the MOS distribution by different zero-shot sampling strategies, as shown in Figure 6, where the sampling times are both 100 ($K = 100$). The orange dots stand for our strategy with various speech prompts, where 10 target transcripts and 10 speech prompts are matched sequentially as zero-shot inputs. The blue dots denote the MOS by Monte-Carlo (MC) Dropout (Gal & Ghahramani, 2016), where 10 transcript-prompts pairs are respectively sampled 10 times with activated Dropout.

From Figure 6, we observe that various speech prompts significantly boost the diversity in generations, thus providing representative training examples with different MOS levels for subsequent optimization. Compared with MC dropout, it can cover more conditions using the same sampling times $K$, thereby potentially reducing the bias in training data collection.

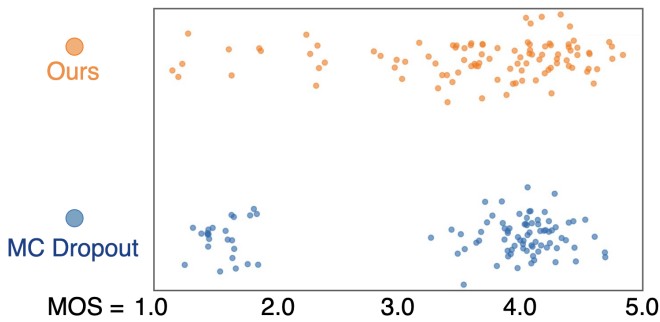

Figure 6: Visualization of sampling strategies.

## D  EXPERIMENTAL DETAILS FOR TRAINING AND EVALUATION

**MOS Data**. To simulate the human annotation, we utilize the SOMOS dataset (Maniati et al., 2022) to train I-CNF and EDL models for uncertainty estimation. SOMOS (full) comprises over 20,000 synthetic speech samples generated by 200 distinct TTS systems. Each audio segment has been assessed by a minimum of 17 different annotators from a pool of 987 participants, with an average of 17.9 annotations per segment.

**I-CNF and EDL**. For both two models, we utilize a frozen WavLM[4] as an upstream backbone and the weighted sum of the outputs of all intermediate Transformer encoder blocks are used as the speech embeddings feeding into the downstream model. The weights are jointly trained with the downstream model which contains two Transformer encoder blocks followed by two fully connected layers. For I-CNF, three real NVP blocks are used for the invertible flow model and 50 samples are drawn for each input speech segment. Theses two model are utilized to label samples with in terms of predicted mean of MOS, as well as providing variance as uncertainty during annotating process.

**MOSNet** [5] is a pre-trained model based on wav2vec2 for MOS prediction (Cooper et al., 2022). Trained from the dataset for VoiceMOS challenge [6], it has good generalization ability on out-of-domain speech assessment. This is the reason we employ it for MOS estimation in UNO experiments. As an evaluator, *MOSNet* is not involved in data selection and annotation.

**UNO training.** We finetune all parameters of pre-trained VoiceCraft models, which are downloaded from Huggingface [7]. Due to the constraint of the reference model, it will not result in over-fitting. The learning rate is set as 1e-5, and the batch size is 2. We employ AdamW as an optimizer and only train for 1 epoch, the training iteration depends on the number of samples. We set sampling times $K$ as 400, and use the *I-CNF* and *EDL* to classify them to 200 of $\mathcal{P}_{pos}$ and 200 of $\mathcal{P}_{neg}$. For 400 samples, it takes around 10 minutes on a single NVIDIA-A100 GPU.

**Human Evaluation.** Our evaluation includes two parts: naturalness MOS and A/B testing. The templates we use to collect feedback from human listeners are presented as follows: 1) Naturalness MOS: *"Please listen to the speech samples and rate how natural each sample sounds in a scale from 1 (very unnatural) to 5 (completely natural), and the scale options are: '1: very unnatural', '2: somewhat unnatural', '3: neither natural nor unnatural', '4: somewhat natural', '5: completely natural'."* 2) A/B testing: *"Please listen to the pairs of speech samples and select the better one for each pair, and the options are: '1: A is better', '2: hard to tell', '3: B is better'."*

## E  EXPERIMENTAL DETAILS FOR HUMAN ANNOTATION

In this experiment, three listeners participate in the annotation and evaluation process. For each batch of 4 zero-shot TTS generations, they are instructed to select 2 desirable and 2 undesirable synthetic

---

[4] https://huggingface.co/microsoft/wavlm-base-plus

[5] https://github.com/nii-yamagishilab/mos-finetune-ssl

[6] https://zenodo.org/records/6572573#.Yphw5y8RprQ

[7] https://huggingface.co/pyp1/VoiceCraft/tree/main

speech as much as possible to balance the positive and negative sample pools. For each sample, we simply record the uncertainty as following rules: (1) If all three listeners consider it desirable, it is placed in the $\mathcal{P}_{\text{pos}}$ with uncertainty set to 0.1. (2) If two individuals consider it desirable, it is also placed in the $\mathcal{P}_{\text{pos}}$ with uncertainty set to 0.5. (3) If only one individual thinks it desirable, then we feed it into $\mathcal{P}_{\text{neg}}$ with an uncertainty of 0.5. (4) If all three listeners consider it undesirable, we feed it into $\mathcal{P}_{\text{neg}}$ with uncertainty of 0.1. With these rules, we finally obtain 216 samples in $\mathcal{P}_{\text{pos}}$ and 184 samples in $\mathcal{P}_{\text{neg}}$. After UNO, these three listeners also evaluate the naturalness of MOS for synthetic speech. Below is the template we use to collect feedback from human listeners: *"Please listen to the batches of speech samples (each batch contains four samples), and select two desirable and two undesirable speech samples."*

## F  EXPERIMENTAL DETAILS FOR EMOTIONAL TTS

**Emotional-State Model** (Mehrabian, 1995) describe human emotions using 3 numerical dimensions: Valence (V), which measures how positive or pleasant emotion ranges from negative to positive; Arousal (A), which measures the agitation level of the person, ranging from non-active / in calm to agitated/ready to act; and Dominance (D) that measures the level of control a person feels of the situation, ranging from submissive / non-control to dominant / in-control. The visualization is shown in Figure 7 sourced from (Deng & Shi, 2022).

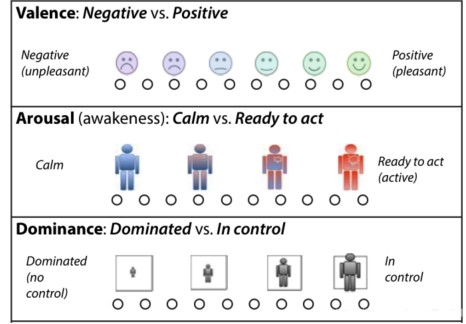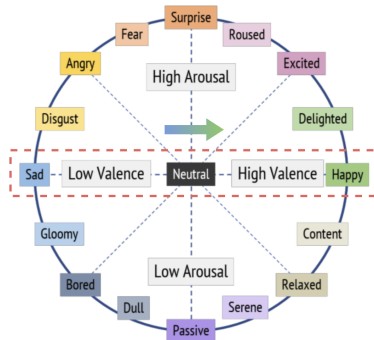

Figure 7: The relationship between Valence-Arousal-Dominance and human emotion

In our experiment 6.5, we aim to increase the valence in synthetic speech as the "blue-to-green" arrow shown in Figure 7. The valence is estimated by a pre-trained neural model [8]. The original TTS model is trained on neutral text-speech pairs, however, the exhibited ability of the zero-shot TTS model shows that the generated speech can mimic the acoustic characteristic of speech prompt. Therefore, we utilize emotional speech as a prompt for zero-shot TTS, thereby encouraging the diversity of valence in synthetic speech. Specifically, we use the same sampling strategy with UNO but replace the speech prompt with "happy" and "sad" categories from emotional ESD dataset (Zhou et al., 2021). The sampling times $K$ contains 1k for happy and 1k for sad, and we first feed top-500 and bottom-500 in terms of valence into $\mathcal{P}_{\text{pos}}$ and $\mathcal{P}_{\text{neg}}$, and then only keep 200 high-quality speech in terms of their MOS. Thereafter, $I$ and $J$ are both 200 and the average valence of $\mathcal{P}_{\text{pos}}$ and $\mathcal{P}_{\text{neg}}$ are respectively 0.65 and 0.36, as shown in Table 3. A similar approach is also applied in arousal experiments to generate speech with surprise emotion. However, since there are no "passive" or "calm" categories in the ESD dataset, we employ "neutral" and contrastive speech prompts for sampling, and the generations with low $a$ and high MOS $m$ are selected to compose $\mathcal{P}_{\text{neg}}$ with $a$ of 0.48.

During the evaluation, our baseline employs Librispeech transcript and pleasant speech prompts with *unseen* speaker during optimization, this is the reason that zero-shot valence is neutral but slightly pleasant (0.55). With the same speech prompt, the model after UNO shows that the valence is significantly improved, where 0.67 is even higher than the average valence in $\mathcal{P}_{\text{pos}}$.

---

[8]`https://github.com/audeering/w2v2-how-to`

## G  VISUALIZATION FOR EVALUATIONS SIMULATED BY I-CNF

To better understand the human annotation simulation, evaluations simulated by I-CNF are visualised against a set of baseline methods including Monte Carlo Dropout (MCDP) (Gal & Ghahramani, 2016), Bayes-by-backprop (BBB) (Blundell et al., 2015), deep ensemble (ENS) (Lakshminarayanan et al., 2017), and conditional variational autoencoder (CVAE) (Kingma & Welling, 2013). Visualisation is shown in Figure 8. The evaluations that have the same score are spread along the $y$ axis according to density to avoid overlapping for visualisation purposes. It can be seen that I-CNF can better match the distribution of scores provided by humans. In contrast, all the other methods tend to either produce annotations centered around the mean score or collapse to one score (typically 3 or 4).

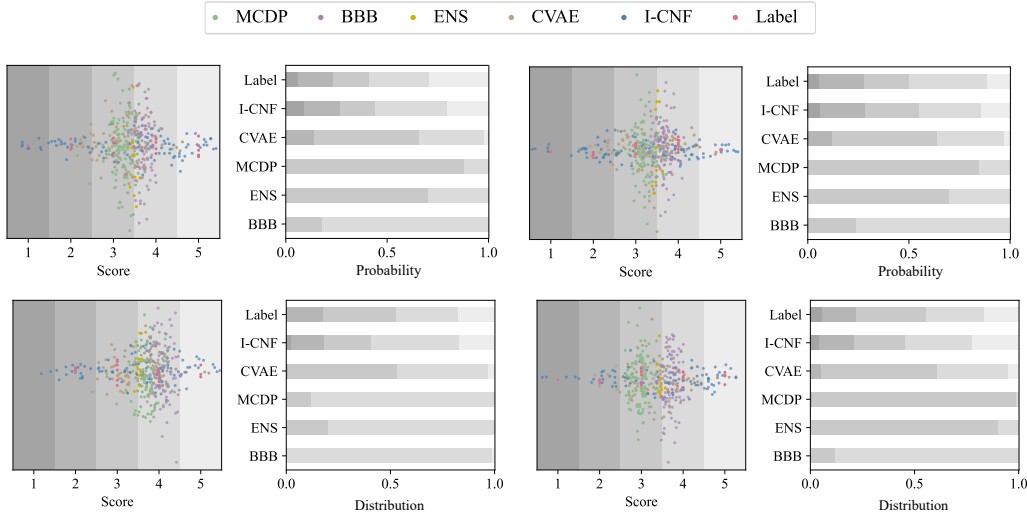

Figure 8: Visualisation of simulated evaluations. For visualisation purposes, the points that have the same scores are spread along the $y$ axis according to density to avoid overlapping.

## H  BROADER IMPACTS

On the positive side, by incorporating human feedback into the training process, the TTS model is improved to provide a better user experience in various applications, such as virtual assistants, accessibility tools for the visually impaired, and language learning platforms. Moreover, this approach can enhance the adaptability of TTS systems to diverse linguistic and cultural contexts, fostering greater inclusivity and accessibility in technology. Overall, the integration of human feedback in TTS optimization can contribute to the development of more sophisticated, user-friendly, and versatile speech synthesis technologies.

Despite the potential benefits, considering that our model demonstrates a high degree of speaker similarity in synthesized speech, it poses potential risks related to misuse, such as spoofing voice identification or impersonating specific individuals. Our experiments were conducted under the premise that the user consents to be the target speaker in speech synthesis. To mitigate these risks, it is imperative to develop a robust synthesized speech detection model and establish a comprehensive system for individuals to report any suspected instances of misuse.

