# OpenReview forum: "Enhancing Zero-shot Text-to-Speech Synthesis with Human Feedback"
_ICLR.cc/2025/Conference — Submitted to ICLR 2025_

### Official Review · Reviewer_Wg5u · 2024-10-21

**Soundness:** 3
**Presentation:** 3
**Contribution:** 2
**Rating:** 5
**Confidence:** 3

**Summary:**

In this paper, the authors introduce an optimization scheme termed Uncertainty-Aware Optimization (UNO) designed to enhance zero-shot Text-to-Speech (TTS) systems through human feedback. Unlike previous RLHF methods, UNO does not require pairwise preference data or a reward model. Experimental results indicate that, following post-training with UNO, the TTS system demonstrates improvements in both subjective metrics (speaker similarity, word error rate) and objective metrics (MOS) compared to the baseline system. Additionally, the authors demonstrate that UNO can seamlessly and flexibly adapt to the desired speaking style in emotional TTS.

**Strengths:**

* This paper introduces uncertainty-aware optimization (UNO), an RLHF method for zero-shot TTS that does not require a reward model or preference data pairs.
* UNO addresses issues in the TTS domain, such as the uncertainty in subjective human evaluations of speech synthesis, offering insights for research in this field.
* The baseline TTS system (VoiceCraft) optimized with UNO shows significant improvements in WER, SIM, and MOS metrics on the LibriSpeech dataset.

**Weaknesses:**

* Firstly, some of the authors' statements are not rigorous. The authors claim that their method does not require a reward model, but in fact, they trained a model to simulate SOMOS and used the results of the SOMOS simulation model as a proxy for the reward, which does not eliminate the need for a reward model.

* Additionally, the authors use SOMOS as a proxy for real human scoring, but the SOMOS samples only include a single speaker (LJSpeech), and its generalization is questionable. I hope the authors will provide more discussion about it.

* The paper shows that the baseline TTS system improves in SIM, WER, and MOS after training with UNO on LibriSpeech, however, the dataset used for UNO training is LibriTTS, which is derived from the original materials of the LibriSpeech corpus, so it is questionable whether the results on LibriSpeech are brought by RLHF. I think it is necessary to check the performance of the model directly fine-tuned on LibriTTS.

* Moreover, the approach of UNO not requiring paired data is not entirely novel; UNO can be seen as REINFORCE with baseline, similar approaches include ReMax [1]  However, I don't think this is a major issue, as UNO may be more suitable for TTS.

* The authors believe that current TTS systems do not exhibit strong diversity in samples generated from the same text and prompt. One possible reason is that the authors tested on LibriTTS, which is an audiobook dataset. In fact, more powerful TTS models (trained on large-scale in-the-wild data) can generate speech with diversity that supports human judgment (at least in some cases) for certain real-life prompts, especially with varying prosody.

Some typos:
* line 215, equation (7): $s_i$ should be $s_j$.
* demo page: "NeurIPS 2024 Conference Submission" should be revised.
* line 815, Table 5: The WER of VALL-E UNO seems to be a typo.

[1] Li Z, Xu T, Zhang Y, et al. Remax: A simple, effective, and efficient method for aligning large language models[J]. arXiv preprint arXiv:2310.10505, 2023.

**Questions:**

Some questions have already been raised in the weaknesses section, with additional questions including:

* The authors use the predicted values of SOMOS as a proxy for the reward, and the experimental results show improvements in the SOMOS metric (which I believe is an expected outcome). I would like to ask the authors if they have tried using WER or SIM metrics directly as proxies for the reward to optimize WER and SIM?

* The authors mention in line 336 that they use WavLM-TDCNN for calculating the SIM metric. However, I believe the model they actually used should be https://huggingface.co/microsoft/wavlm-base-sv instead of WavLM-TDCNN, as the results on LibriSpeech generally exceed 0.8. As far as I know, the current SOTA models using WavLM-TDCNN for calculating the SIM metric typically yield results around 0.7.

* Have the authors explored the generalization capabilities of their model on in-the-wild data?

---

> ### Author Response · Authors · 2024-11-17
> **Response to Reviewer Wg5u (1/2)**
>
> Thank you for the time and effort you've put into reviewing. Based on your valuable feedback, we have summarized the following issues and responded to each one to address your concerns.
>
> Q1: Proxy model vs. reward model. \
> Thank you for your question. Let us first clarify the role of the simulation proxy model: its primary function is to provide a binary classification of "desirable" or "undesirable." Ideally, this task would be carried out by human annotators, with uncertainty represented by the variance among their judgments. However, due to the substantial labor demands, we used either EDL or I-CNF as a proxy. As noted by the reviewer, current MOS prediction models often struggle with generalization when applied to newly developed TTS systems, as the publicly available MOS prediction datasets are typically based on older, more traditional TTS models. Consequently, we did not use these models as reward models for generating precise numerical values. Instead, we applied a threshold to convert their outputs into binary classifications. In this capacity, they are reliable; for instance, synthesized speech with a MOS score below 3.0 is very likely a poor case, while scores above 4.5 generally indicate near-perfect quality.
>
> Q2: Generalization of the proxy model. \
> As we explained in Q1, the proxy model is not accurate but suitable to be a binary classifier. We also have human listeners to determine "desirable" or "undesirable" in Table 2, which shows the efficacy of the proxy model.
>
> Q3: Whether the improvements come from using the LibriSpeech dataset or RLHF? \
> Thanks for your question. We believe the improvement comes from RLHF post-training. The reason is that we did not use any “labels” (which are ground-truth speech audios here) from the LibriSpeech dataset. Actually, RLHF finetuning learns from human feedback instead of labels, which is its key difference from SFT.  \
> Furthermore, the 830M model can be viewed as the STF baseline, because it adds part of the LibriTTS model in its training set (https://huggingface.co/pyp1/VoiceCraft_830M_TTSEnhanced), which archives 4.30 zero-shot performance. However, UNO and UNO-RI improved it to 4.41 and 4.55, respectively.
>
> Q4: Novelty of UNO on TTS task. \
> Thank you for your suggestion. We agree with your perspective that UNO can be viewed as a TTS-specific adaptation RLHF method. In particular, we have carefully designed the methodology to enhance sampling efficiency by exposing the model to a greater diversity of speakers, which significantly improves the robustness and performance of the SFT TTS model. We believe our contributions are specifically tailored to the TTS domain, providing sufficient novelty in this context.
>
> Q5: Powerful TTS models exhibit strong diversity in synthesized speech. \
> We appreciate the reviewer’s insightful question and would like to clarify two distinct meanings of the term “diversity” in this context. State-of-the-art zero-shot TTS models can generate natural and diverse speech by mimicking a speaker's voice characteristics and speaking style based solely on the input speech prompt (typically the first three seconds of speech). The diversity here in the generated speech arises from the evidence provided in the speech prompt, which can be expressive enough to convey emotional speech. However, this type of diversity is different from the diversity involved in sampling pairwise positive-negative examples based on the same speech prompt. \
> In practice, we observed that for a given input speech prompt, the generated samples tend to be either all positive or all negative, with only minor prosody variations. This is because, given the prosody in the initial three seconds and the text to be spoken, the pronunciation and prosody of the subsequent speech are generally predictable. Consequently, it is rare to find training samples that share the same speech prompt and text inputs but exhibit significantly different pronunciations or prosodies.

---

> ### Author Response · Authors · 2024-11-17
> **Response to Reviewer Wg5u (2/2)**
>
> Q6: Different proxy for rewards to optimize the WER and SIM. \
> Thank you for your question. Yes, recognizing the importance of positive-negative selection, we have experimented with various selection rules, including WER and SIM. The results indicate that intelligibility and quality are highly positively correlated in VOICECRAFT's synthesized speech. Specifically, cases with MOS below 3 tend to have high WER, while samples with high MOS typically exhibit low WER. However, in practice, using WER+MOS or SIM+MOS as selection criteria did not result in significant performance improvements.
> That said, once quality is ensured, the proxy can be customized based on specific objectives. For example, in our results on emotional TTS, we employed valence or arousal metrics as proxies to optimize the emotional expressiveness of the synthesized speech.
>
> Q7: Generalization to in-the-wild data. \
> Thanks for your question. The last 8 samples shown on our demo page (https://uno-tts.github.io/listening-examples/) are the emotional TTS experiments generated from totally unseen speakers with unseen emotions (from ESD[1]). UNO can improve the Valence+MOS and Arousal+MOS.
>
> Reference \
> [1]Zhou K, Sisman B, Liu R, et al. Seen and unseen emotional style transfer for voice conversion with a new emotional speech dataset[C]//ICASSP 2021-2021 IEEE International Conference on Acoustics, Speech and Signal Processing (ICASSP). IEEE, 2021: 920-924.

---

### Official Review · Reviewer_BvGc · 2024-10-27

**Soundness:** 3
**Presentation:** 2
**Contribution:** 3
**Rating:** 6
**Confidence:** 3

**Summary:**

This paper introduces a novel framework, UNO, aimed at integrating human feedback into the text-to-speech (TTS) learning loop. The proposed framework leverages an uncertainty-aware optimization approach and follows a sampling-annotating-learning pipeline. First, the authors generate multiple samples by using inference with varied prompts. Next, they train an annotation simulator to model human decision-making with associated uncertainties. Finally, the framework fine-tunes the pre-trained model through a proposed Uncertainty-aware Optimization loss.

**Strengths:**

1. This work addresses an important problem by incorporating human feedback into TTS model training, and it is theoretically well-founded.
2. The authors present extensive evaluations and experiments, providing a thorough analysis of the proposed framework.
3. The framework appears to be a generalizable pipeline, though additional analyses and experiments would further strengthen the work.

**Weaknesses:**

1. In the simulation and annotation steps, the simulators I-CNF and EDL rely on systems that are already well-established in prior research, limiting the novelty of the annotation procedure.
2. The equations in the paper could benefit from better formatting, as double references, such as "Eqn. equation," detract from readability.
3. Many important points and comparisons are located in the appendix. These should be moved to the main sections to improve the readability. The writing should be improved.

**Questions:**

1. In lines 282-283, the authors mention that due to the lack of diversity, existing TTS models struggle to generate paired sw and sl using fixed target transcripts and speech prompts. I am unclear about this assumption, as many TTS models can produce varied speech by using different random seeds in the sampling process. Could you clarify this point?
2. Why does VALLE-UNO perform significantly worse than VALLE on the WER metric? Given the high WER, it’s surprising that both similarity and MOS scores remain high.
3. In line 794, author mentioned that theoretically feasible for UNO to optimize diffusion-based TTS. Could you provide experimental results to support UNO’s application to other diffusion-based TTS models?
4. On the demo page, the authors highlight some failure cases for the Voicecraft system. Assuming similar issues could arise after UNO optimization, could you provide comparative statistics on these failure rates?

---

> ### Author Response · Authors · 2024-11-17
> **Response to Reviewer BvGc**
>
> Thank you for the time and effort you've put into reviewing. Based on your valuable feedback, we have summarized the following issues and responded to each one to address your concerns.
>
> Q1: The diversity of the zero-shot TTS system. \
> We appreciate the reviewer’s insightful question and would like to clarify two distinct meanings of the term “diversity” in this context. State-of-the-art zero-shot TTS models can generate natural and diverse speech by mimicking a speaker's voice characteristics and speaking style based solely on the input speech prompt (typically the first three seconds of speech). The diversity here in the generated speech arises from the evidence provided in the speech prompt, which can be expressive enough to convey emotional speech. However, this type of diversity is different from the diversity involved in sampling pairwise positive-negative examples based on the same speech prompt.
> In practice, we observed that for a given input speech prompt, the generated samples tend to be either all positive or all negative, with only minor prosody variations. This is because, given the prosody in the initial three seconds and the text to be spoken, the pronunciation and prosody of the subsequent speech are generally predictable. Consequently, it is rare to find training samples that share the same speech prompt and text inputs but exhibit significantly different pronunciations or prosodies.
>
> Q2: Q2: WER of VALL-E-UNO. \
> We apologize for any misunderstanding caused. Here is a typo and the real result should be 15.8 rather than 157.8, and we will correct it in the next version.
>
> Q3: Could you provide experimental results to support UNO’s application to other diffusion-based TTS models? \
> We thank the reviewer for the suggestion. As stated in the limitation section, we believe UNO may be feasible for TTS models with diffusion. However, current zero-shot TTS models have mostly used the autoregressive approach, which is often not yet widely combined with diffusion. Moreover, UNO, as a reinforcement learning approach, benefits from alleviating the exposure bias issue caused by error propagation in autoregressive decoding. Current TTS methods with diffusion are mostly non-autoregressive, which may not suffer as many instability issues as the autoregressive ones.
>
> Q4: Provide comparative statistics on these failure rates? \
> Thanks for your question. As we reported in Section 6: If we define WER>15% as “failure rates”, then the 330M drops from 18.5% to 4.9%, and the 830M mode drops from 4.6% to 1.9%. With UNO-RI, this figuare reduce to less than 1%. A similar reduction is also observed if we use the MOS value as the threshold. However, due to the lower unreliability of MOS estimation, we select WER as this threshold.
>
>  We would be delighted to receive any additional suggestions or comments.

---

> > ### Comment · Reviewer_BvGc · 2024-11-22
> >
> > Thanks for your kind response. I will keep my score, but still look forward to your next version.

---

### Official Review · Reviewer_782F · 2024-10-28

**Soundness:** 2
**Presentation:** 2
**Contribution:** 2
**Rating:** 3
**Confidence:** 4

**Summary:**

The paper explores integrating subjective human feedback into the training process of text-to-speech (TTS) models through a method called uncertainty-aware optimization (UNO). It aims to address the mismatch between TTS training and evaluation methods like the mean opinion score (MOS). UNO introduces a framework that incorporates human evaluation into training, using a sampling-annotating-learning approach. It focuses on optimizing TTS performance without needing preference data, accommodating the inherent variability in human feedback. The method claims to improve zero-shot TTS performance in speaker similarity, word error rate, and emotional speech synthesis.

**Strengths:**

**Integration of Human Feedback** This paper introduces an interesting approach by incorporating subjective human evaluation directly into the TTS training process through the Uncertainty-Aware Optimization (UNO) method. This aims to address a key challenge in aligning TTS models with human preferences and eliminates the need for complex reward models or preference data.

**Flexible and Adaptable Method**  The UNO method is presented to be adjust to different speaking styles, including emotional TTS, offering flexibility in generating synthetic speech.

**Weaknesses:**

**Limited Novelty of the Proposed Approach** The UNO method presents an algorithm that closely resembles the KTO algorithm, which was introduced earlier on February 2, 2024. While UNO incorporates uncertainty into its framework, the results in Section 6.3 indicate that it achieves comparable Mean Opinion Scores (MOS) to the UNO-null variant (which does not have uncertainty), with scores of 4.31+0.66 and 4.24+0.59, respectively. This suggests that the inclusion of uncertainty does not significantly enhance performance compared to KTO. Given this similarity and the limited improvement of uncertainty, the novelty of the proposed approach appears to be minimal, as the core methodologies are not fundamentally new. This raises questions about the contributions of UNO to the field, as it does not seem to offer substantial advancements over existing techniques. Can you provide a more detailed comparison between UNO and KTO, highlighting any key differences or improvements? Additionally, can you elaborate on the specific contributions of incorporating uncertainty, given the similar performance to the UNO-null variant?

**Potential limitation in the paper's approach to aligning TTS training with MOS-like subjective evaluation metrics** In introduction, you argue that the TTS system has a clear mismatch between the training objectives and human evaluation. Motivated by this, you propose to use UNO to incorporate human feedback into the TTS training process, aiming to align training objectives with MOS-like subjective evaluation metrics. You train I-CNF with real human-labeled SOMOS dataset to simulate human-like annotations. However, because SOMOS mainly evaluates naturalness, a TTS model optimized for a high naturalness score might still struggle with intelligibility, leading to possible issues in accurately mapping the generated speech to the text content.

**Evaluation discrepancies for Voicecraft** Voicecraft is reported to achieve high MOS for both intelligibility (MOS of 4.23±0.06) and naturalness (MOS of 4.17±0.06) based on evaluations by 59 human participants in the original paper. But you report significantly lower MOS scores for Voicecraft (3.65 by MOSNet and 3.38 from 10 participants) in the evaluation results. This raises concerns about the reliability of the evaluation. If it is because you use LibriTTS to evaluate Voicecraft but the original paper use REALEDIT dataset, it is unclear why you did not utilize REALEDIT, which is publicly available, for fine-tuning with UNO, which could strengthen the comparison.

**Lack of clarity on the accuracy of EDL and I-CNF in predicting human-like annotations and their associated uncertainty** Although the paper compares the distribution of I-CNF predictions and human labels, it doesn’t demonstrate the accuracy and reliability of these models. Given that these predictions significantly impact UNO's overall performance, without validation of EDL and I-CNF's accuracy, it’s challenging to assess the trustworthiness of the results and the proposed method’s effectiveness.

**Validity of ICNF as Human Feedback Proxy** In table 2, the UNO-Human experiment has lower human MOS and MOSNet MOS compared to UNO-ICNF seem unusual and raises concerns. This could suggest that the ICNF predictions might not be truly representative of human preferences. It raises questions about the validity of ICNF as a proxy for human feedback—if real human feedback leads to poorer performance, it may indicate that the ICNF model is capturing a different aspect of the data that is not entirely aligned with human judgment. The paper should clarify why ICNF annotations outperform human-labeled data and whether the evaluation method favors ICNF annotation over true human annotation.

**Questions:**

1. In Section 4.1, you mention distinguishing desirable and undesirable samples through human feedback, stored in separate pools (P_pos and P_neg). However, in the third paragraph, you state that EDL and I-CNF are trained on the SOMOS dataset to simulate human-like annotations and uncertainty, rather than through true human distinction. Could you clarify how "human-like" annotations from EDL and I-CNF are predicted? Does I-CNF predict a MOS score per audio, given that SOMOS only provides MOS scores? If EDL and I-CNF are trained to predict MOS with uncertainty, the method for classifying samples into desirable and undesirable categories based on these predictions remains unclear. Can you provide a detailed explanation of the process for classifying samples into P_pos and P_neg, including any specific thresholds used and how these were determined? Can you provide a sensitivity analysis of the boundary score? This gap in the methodology makes it difficult to understand the criteria for categorizing samples into P_pos and P_neg​.

2.  Can you provide confidence intervals or p-values for all key comparisons, particularly those between UNO and the baseline methods (PPO, DPO, and ODPO) across all reported metrics? Without these, it is impossible to tell if the result is significantly better than PPO and DPO and ODPO.

3. The claim in Appendix D, UNO training: “Due to the constraint of the reference model, it will not result in over-fitting.” is not correct.

4. Can you provide qualitative examples or audio samples that illustrate the improvements made by your approach?

5. The experiment in 6.5 EXTENSION ON EMOTIONAL TTS and Appendix F is not clear. How do you identify the samples with high and low valence and also arousal, do you train a model to identify them? What is your evaluation metric for identifying the emotion state of UNO's results, the same model you used for identifying high and low valence/arousal? Can you also provide some audio samples that illustrate the emotion change made by your approach?

**Details Of Ethics Concerns:**

Research involving human subjects

---

> ### Author Response · Authors · 2024-11-17
> **Response to Reviewer 782F (1/2)**
>
> Thank you for the time and effort you've put into reviewing. Based on your valuable feedback, we have summarized the following issues and responded to each one to address your concerns.
>
>
> Q1: Question about EDL and I-CNF, and the selection of positive and negative examples. \
> In short, EDL and I-CNF predict a MOS distribution (mean and variance). The SOMOS dataset includes the individual MOS scores from each listener, which supports these models in predicting the distribution. Regarding the positive and negative selection, we provide the following observations and discussions to address your concern: \
> (1) Although the predicted means of EDL and I-CNF may not serve as accurate MOS values due to the mismatch between their training data, produced by older TTS systems, and the outputs of recent TTS systems, we found that they are well-suited to function as binary classifiers for identifying positive or negative samples. For instance, when the predicted MOS is below 3.0, the synthesized speech usually has various issues, whereas samples with a score above 4.5 generally sound nearly perfect. Therefore, we use them to automatically identify negative samples.  \
> (2)  In the experiments of the 330M model, we set thresholds of 3.5 and 4.0 to determine whether a sample was “desirable” or “undesirable”. For samples whose MOS scores are from 3.5 to 4.0, EDL and I-CNF may not be able to provide meaningful differentiation. Fortunately, the optimization algorithm is not sensitive to these intermediate samples—they are usually observed relateive high uncertainty to limit the model update. Therefore, whether they are included in the positive or negative sample pool has little impact on the final outcome.   \
> (3) In the 830M-model experiment, we found that the synthesized speech quality was generally very high. Therefore, we raised the two thresholds to 3.9 and 4.4. Under these conditions, we observed that uncertainty could provide additional differentiation. Specifically, we aim for the model to optimize towards high MOS scores while maintaining low uncertainty. Therefore, UNO surpasses the UNO-null due to the contribution of uncertainty. \
> (4) In Section 6.4, we further explored a method for refining the sample selection when a large number of samples are available—namely, reverse inference. Using this approach, we could further identify the truly "desirable" examples, ultimately resulting in improved outcomes of 830M model.
> In general, the selection criterion of  "desirable" and “undesirable” is flexible: For the 330M model, it’s important to **filter out those bad cases** to improve the robustness. For the 830M model, **more strict rules** are needed for selecting real "desirable" samples.
>
>
> Q2: The question about emotional TTS and listening examples. \
> The model we used for the prediction of valence and arousal is in the footnote of page 19: (https://github.com/audeering/w2v2-how-to). As we mentioned, when the 830M model can synthesize speech with high MOS, the emotional TTS actually adds more rules for sample selection—we want the model to be optimized toward the direction with high MOS and high valence (or arousal). The motivation of experiments is to show the flexibility of our optimization strategy. When the quality issue is solved, UNO can also be used for preference optimization.  Please find the listening examples on our demo page in the footnote of page 8 (https://uno-tts.github.io/listening-examples/).
>
>
> Q3:  Evaluation discrepancies for Voicecraft. \
> We would like to clarify the confusion here. The 4.17 MOS reported in VoiceCraft is evaluated with their 830M model. Our main experiments were conducted using the 330M model, and we also have the experiments on the 830M model, the MOS score of zero-shot synthesis is 4.30, even better than the number reported in the original paper. Additionally, the 830M model actually is enhanced by a part of LibriTTS, while the 330M model we used is only trained by Gigaspeech. Therefore, the proposed method handles the slight domain mismatch and unseen speakers, achieving similar performance with the 830M model without any ground-truth speech on LibriTTS.

---

> ### Author Response · Authors · 2024-11-17
> **Response to Reviewer 782F (2/2)**
>
> Q4: Why does real human feedback lead to poorer performance?  \
> This phenomenon occurs because, unlike the fixed threshold used for the EDL model, we could not establish a consistent threshold for human raters. Instead, listeners were instructed to select two desirable and two undesirable samples from a batch of four, which resulted in less effective selection. Furthermore, since each audio sample was evaluated by only three listeners, the uncertainty calculation was relatively imprecise, as explained in Appendix E. \
> Additionally, we found that sometimes the model itself is better suited as a judge of synthesized speech than human listeners. As we did in Section 6.4, when people were unable to determine the quality difference between two synthesized speech samples, using the TTS model itself for reverse inference often led to selecting better positive samples.
>
> Q5: Clarity on the accuracy of EDL and I-CNF.  \
> For mean MOS prediction, we have highlighted that no model currently provides accurate MOS values due to the rapid progress made in developing TTS systems. Therefore, we use these models as binary classifiers to filter representative examples. Regarding uncertainty, we visualized a comparison of I-CNF's uncertainty predictions with other methods in Appendix G. Combined with the metrics presented in the main text, this evidence demonstrates that I-CNF is the state-of-the-art method for estimating MOS uncertainty.
>
>
> Q6:  A TTS model optimized for a high naturalness score might still struggle with intelligibility, leading to possible issues in accurately mapping the generated speech to the text content. \
> We apologize for any misunderstanding caused. In VoiceCraft's synthesized speech, intelligibility and quality are **highly positively** correlated. Samples with a MOS below 3 typically exhibit high WER, while those with high MOS generally have low WER. The reviewer can refer to our listening demo page, where samples with improved quality do not show a decline in intelligibility. Furthermore, during our initial explorations, both WER and SIM scores were considered for sample selection. However, SIM was found to be unreliable, and combining WER with MOS as a standard did not result in significant performance improvements.
>
> Q7: Highlighting any key differences or improvements between UNO and KTO, and the contributions of incorporating uncertainty. \
> As stated in line 390, our approach is inspired by KTO—a method that does not rely on pairwise comparisons when designing the reference point. Using a reference point has been a common approach in reinforcement learning optimization since its introduction in the 1990s [1]. Even in recent years, for optimizing sequence generation, subtracting a reference point from the reward to stabilize training remains a standard practice [2]. Additionally, the differences between our method and KTO include:
> - KTO primarily focuses on addressing the bias inherent in human feedback, particularly the well-documented tendency for humans to be loss-averse. To account for this, they incorporate a loss aversion coefficient to model this aspect of prospect theory. However, this approach was not effective in our experiments, leading us to omit this setting.
> - UNO set beta is related to the uncertainty of each data point, and the importance of dynamic beta is also explored in paper [3]. The first advantage of a dynamic beta is filtering to safeguard against the influence of outliers. As the EDL and I-CNF models provide a rough judgment of whether a sample is desirable or undesirable, if an ambiguous sample (typically characterized by high uncertainty) appears, its high uncertainty prevents the model from over-adjusting towards this outlier during the learning process.
> - The second advantage is shown in the comparison with UNO-null. Though UNO-null works well with the 330M model, however, it fails to improve the performance of the 830M model (Figure 4). This is mainly due to the 830M model's ability to produce relatively stable sampling results. (Typo here) Both EDL and I-CNF could provide sufficient differentiation in these cases. However, a dynamic uncertainty metric allows for further quality improvement.
>
> We hope that the above discussion can clarify the reviewer's misunderstandings and address proposed concerns. We would be delighted to receive any additional suggestions or comments.
>
>
> Reference  \
> [1] Ronald J. Williams. Simple statistical gradient-following algorithms for connectionist reinforcement learning. In Machine Learning, pages 229–256, 1992.\
> [2] Rennie S J, Marcheret E, Mroueh Y, et al. Self-critical sequence training for image captioning[C]//Proceedings of the IEEE conference on computer vision and pattern recognition. 2017: 7008-7024. \
> [3] Wu J, Xie Y, Yang Z, et al. $\beta$-DPO: Direct Preference Optimization with Dynamic $\beta$ [J]. arXiv preprint arXiv:2407.08639, 2024.

---

> ### Comment · Reviewer_782F · 2024-11-21
>
> Thank you for your response. However, it does not provide the accuracy of EDL and I-CNF.  I understand that there may be some mismatch between their training data and your synthesized data, but this is the potential issue of whether EDL and I-CNF can correctly predict human-like annotations and human’s associated uncertainty of your synthesized data. Furthermore, as you noted, the EDL model relies on a fixed threshold, whereas human listeners do not. We don’t know the accuracy of EDL and I-CNF on predicting human feedback and your UNO-Human experiment demonstrates that the EDL model behaves differently from human listeners. it is questionable whether UNO truly enhances zero-shot TTS with human feedback as you claimed.
>
> Additionally, I still have questions regarding the contribution of your work. As you noted, your method is inspired by KTO and is quite similar to it. One key difference is the inclusion of uncertainty, achieved by setting a beta parameter related to the uncertainty of each data point. However, your experiments do not clearly demonstrate whether the inclusion of uncertainty significantly enhances performance compared to a method without uncertainty:
>  1. UNO-null can improve the 330M model just like UNO-ICNF
>  2. For the 830M model, neither UNO-null nor UNO-ICNF results in a significant performance improvement (as shown in Figure4)
>  3. You state that neither EDL nor I-CNF provides sufficient differentiation in these cases.
>
> Given these observations, how can you confidently claim that the inclusion of uncertainty significantly enhances performance? What is the contribution of your method?

---

> ### Author Response · Authors · 2024-11-21
> **Resoponse to Reviewer 782F**
>
> Thank you for your response. We would clarify that The EDL and I-CNF serve as **Proxy model** to reduce dependence on human resources. In this case, the criterion for determining whether they are qualified lies in whether they can improve the optimization performane, and the comparative exprement with real annotator also validates their effctiveness. The threshold they use is because as neural networks, they have to use scores to show the preferences, unlike annotator can directly indicate which one is better. But there is essentially no difference between these two  judgment methods. For example, if we employ a language model as a judgment model to perform online perference optimization. Then, if this LM can achieve similar (even better) results as real annotator, we believe it is a qualified proxy, and it is NOT impotant whether LM use a socre or directly indicate a candidate.
>
> We apologize for the misunderstandings caused by "**neither**". What we intended to declare was that they could provide sufficient differentiation (sorry for the typo in the last round response). For 830M model, since the sampling results are good enough and the uncertainty can provide further differentiation for these samples. We also highlight that though 4.30 to 4.41 is not a significant MOS improvement, the bad case ratio (WER>15%)  drops from 4.6% to 1.9%, further improving the robustness.
>
> Ours main contirbution is to propose a TTS-specific adaptation RLHF method. In particular, we designed the strategy to enhance sampling efficiency by exposing the model to a greater diversity of speakers. As KTO's data format support the setting of this sampling strategy, we remove some ineffective KTO components and introduece the TTS-specific uncertanty, to make the new method effective on different models.
>
> Thank you again for joining the discussion. We would be delighted to receive any additional suggestions or comments.

---

> > ### Comment · Reviewer_782F · 2024-11-25
> >
> > Thank you for providing more information. If EDL and ICNF does not need to accurately align with human feedback  as long as EDL and ICNF can improve the optimization performance. Then probably you should change the paper as Enhancing Zero-shot Text-to-Speech Synthesis with EDL and ICNF since we don’t know whether human feedback are actually involved or not and how much it is included.
> >
> > Can you provide confidence intervals or p values for UNO-null and UNO-ICNF? And what is the bad case ratio (WER>15%) WER for UNO-null?
> >
> > Also this question in my initial review is not answered, can you please provide some explanation?
> > Can you provide confidence intervals or p-values for all key comparisons, particularly those between UNO and the baseline methods (PPO, DPO, and ODPO) across all reported metrics? Without these, it is impossible to tell if the result is significantly better than PPO and DPO and ODPO.

---

> > > ### Author Response · Authors · 2024-11-26
> > > **Response to Reviewer 782F**
> > >
> > > Thanks for your response.
> > >
> > > For the EDL and ICNF, we would further clarify that: (1) These two proxy models are also trained using human annotated data. (2) We involved the **real human feedback** experiment in Table 2, which replaces the proxy models to provide preference and uncertainty.
> > >
> > > For the bad case ratio of UNO-null, the 830M model is 3.7% (UNO-ICNF 1.9%), and the 330M model is 5.2% (UNO-ICNF 4.9%). Moreover, as we metioned, UNO-ICNF are not sensitive to the data imbalance (postive-negative ratio) due to uncertainty, as those ambiguous samples are assigned high uncertainty to constrain the update step. Specifically, (1) When this ratio is 1:4, the MOS of UNO-ICNF is 4.20, while the UNO-null is 3.98. (2) When the ratio is 4:1, the MOS of UNO-ICNF is 4.01, while the UNO-null is only 3.77 (330M model). Since we use proxy model, we do not highlight this advantage since data imbalance can be easily addressed by increasing sampling times. However, when preference is given by human, the optimization algorithm should be able to handle silght data imbalance issue, as the desirable and undesirable examples may not always be 1:1.
> > >
> > > We appriacte your advice on calculating p-value. For UNO-ICNF and UNO-null, the p-value of MOS model is $1.13\times e^{-5}$ (830M model) and $3.97 \times e^{-3}$ (330M model).  The p-values of PPO, DPO, and DDPO are actually very small. we report the p-values of baselines in the following tables.
> > > | Baseline   |  MOS [1-5] | WER [0-1] | SIM (0-1) |
> > > |------------|--------------|------------|------------|
> > > PPO   |      $1.78 \times 10^{-31}$ |  $4.91\times 10^{-12}$   |   $2.74\times 10^{-3}$    |
> > > DPO   |     $1.24 \times 10^{-40}$  |   $8.25\times 10^{-9}$  |    $5.30\times 10^{-2}$    |
> > > ODPO |      $1.69 \times 10^{-35}$  |   $3.32\times 10^{-8}$  |    $1.02\times 10^{-2}$   |
> > >
> > >
> > > We hope the reponse can address your concerns, and thank you again for joining the discussion.

---

> ### Author Response · Authors · 2024-12-02
> **Response to Reviewer 782F**
>
> Thank you once again for your valuable feedback. We believe that our previous responses have addressed your concerns and clarified some misunderstandings. As the discussion phase is coming to a close, we kindly request if you would consider adjusting your rating. At the same time, we welcome any further comments or suggestions you may have, and we will do our best to address them before the conclusion of the discussion phase.

---

### Official Review · Reviewer_DNwS · 2024-10-31

**Soundness:** 4
**Presentation:** 3
**Contribution:** 4
**Rating:** 8
**Confidence:** 3

**Summary:**

This paper introduces a novel optimization framework, UNO, designed for zero-shot TTS systems, which integrates human feedback with the original TTS objectives. The proposed method enhances a TTS model through a three-step process:
1. Generating hundreds of samples using diverse speech prompts.
2. Annotating these samples with binary labels (like or dislike) based on feedback from real human raters or simulated by neural networks.
3. Training the TTS model using a combination of the original TTS objective and the (pseudo-)human feedback objective.
To account for the uncertainty and confidence associated with pseudo-human feedback, the framework incorporates an uncertainty-aware optimization strategy. This approach encourages the model to update more aggressively when presented with low-uncertainty samples and more conservatively with high-uncertainty ones.

In contrast to previous methods like DPO, UNO eliminates the need for paired good/bad samples, simplifying the sample generation process outlined in step 1. Experimental results demonstrate that UNO effectively improves correctness (as measured by WER), speaker similarity, and overall speech quality (measured by MOS) across TTS models.

**Strengths:**

- The paper is well-written, presenting a clear logical flow and strong motivation, supported by sufficient background knowledge. The authors effectively present the rationale behind their solution and provide intuitive explanations for the mathematical formulation, which might initially seem complex. The differences and improvements of the proposed method compared to previous work are also clearly outlined.
- The UNO framework consistently shows advantages when applied to various TTS models.
- A set of ablation studies is conducted to validate the rationale and effectiveness of the proposed method, including simulations of human feedback using a neural network (as seen in Table 2) and the significance of uncertainty-aware optimization (demonstrated in Table 4), etc.
- The emotional TTS experiments indicate that the UNO framework can be adapted to any binary signals, extending beyond human preferences to emotional valence and arousal, and potentially more factors.
- The Q&A section in the appendix candidly addresses the limitations of this work and outlines potential future directions for further research in this area.

**Weaknesses:**

On page 5, the definitions of EDL and I-CNF are clear but somewhat overwhelming. I recommend a minor restructuring of these sections to enhance readability and make the content easier to follow.

**Questions:**

On Table 5 of page 16, the performance of VALL-E-UNO is unreasonably bad. The authors may need to further check the numerical results in their tables.

---

> ### Author Response · Authors · 2024-11-17
> **Response to Reviewer DNwS**
>
> Thank you for the time and effort you've put into reviewing. Based on your valuable feedback, we have summarized the following issues and responded to each one to address your concerns.
>
> Q1: The performance of VALL-E-UNO is unreasonably bad.  \
> Thank you for pointing out this typo. The correct result should be 15.8, not 157.8, we will correct it in the next version.
>
> Q2:  the definitions of EDL and I-CNF. \
> Thanks for your advice, we are willing to include more details to improve its readability.

---

> > ### Comment · Reviewer_DNwS · 2024-11-20
> >
> > Thanks! Don't forget to also check the results in other tables agin.

---

### Meta-Review · Area_Chair_UdTq · 2024-12-20

**Metareview:**

This paper explores integrating subjective human feedback into the training process of text-to-speech (TTS) models through a method called uncertainty-aware optimization (UNO). It aims to address the mismatch between TTS training criteria and evaluation metrics like the mean opinion score (MOS). UNO introduces a framework that incorporates human evaluation into training, using a sampling-annotating-learning approach, similar to KTO. It focuses on optimizing TTS performance without needing preference data (paired samples) or a reward model, and accommodates the inherent variability in human feedback. Authors claim UNO improves zero-shot TTS performance in speaker similarity, word error rate, and emotional speech synthesis (MOS), so with both subjective and objective criteria.

Strengths
- the paper treats an interesting and relevant subject, if shown to generalize
- authors and reviewers engaged in an in-depth discussion
- the paper is generally well written

Weaknesses
- After discussion, reviewers are still unconvinced that the proposed method (especially wrt uncertainty incorporation) outperforms the baseline, since the observed gains could also be due to fine-tuning on test data (LibriSpeech)
- Improvements due to incorporation of uncertainty are also unclear after the author response
- Confidence intervals seem to be almost unreasonably narrow in the authors' formulation, the interpretation of results is overall not clear and convincing

Overall, given the open questions around how to interpret the results, I am not recommending acceptance at this point.

**Additional Comments On Reviewer Discussion:**

Two eviewers asked several clarification questions to the authors and iterated on their understanding. These two reviewers do not recommend acceptance and also did not raise their score, their questions are listed as weaknesses above.

---

### Decision · Program_Chairs · 2025-01-22

Reject